# Identification and Functional Characterization of Novel MYC-Regulated Long Noncoding RNAs in Group 3 Medulloblastoma

**DOI:** 10.3390/cancers13153853

**Published:** 2021-07-30

**Authors:** Jessica Rea, Annamaria Carissimo, Daniela Trisciuoglio, Barbara Illi, Daniel Picard, Marc Remke, Pietro Laneve, Elisa Caffarelli

**Affiliations:** 1Department of Biology and Biotechnologies “C. Darwin”, Sapienza University of Rome, 00185 Rome, Italy; jessica.rea@uniroma1.it; 2Institute for Applied Mathematics “Mauro Picone”, CNR, 80131 Naples, Italy; a.carissimo@na.iac.cnr.it; 3Institute of Molecular Biology and Pathology, CNR, 00185 Rome, Italy; daniela.trisciuoglio@cnr.it (D.T.); barbara.illi@cnr.it (B.I.); 4Department of Pediatric Oncology, Hematology and Clinical Immunology, Medical Faculty, University Hospital Düsseldorf, 40225 Düsseldorf, Germany; daniel.picard@med.uni-duesseldorf.de (D.P.); Marc.Remke@med.uni-duesseldorf.de (M.R.); 5Department of Neuropathology, Faculty of Medicine, University Hospital Düsseldorf, 40225 Düsseldorf, Germany; 6Division of Pediatric Neuro-Oncogenomics, German Cancer Research Center (DKFZ), 69120 Heidelberg, Germany; 7German Consortium Neuro-Oncogenomics Cancer Research (DKTK), Partner Site Essen/Düsseldorf, 40225 Düsseldorf, Germany

**Keywords:** medulloblastoma, group 3, MYC, OMOMYC, oncogene, RNA-Seq, long noncoding RNAs, apoptosis, migration

## Abstract

**Simple Summary:**

Medulloblastoma is the most common malignant pediatric brain tumor, which accounts for approximately 20% of all childhood brain tumors. To date, no pharmacological approaches are decisive in the treatment of this cancer, while the secondary effects of conventional therapies as chemotherapy, radiotherapy or surgical interventions heavily affect the quality of life of patients. This requires the rapid development of alternative molecular therapies, which are the future challenge of personalized medicine. In this context, we addressed our research towards the most aggressive form of Medulloblastoma to identify novel genes responsible for its onset and/or progression. We discovered three newly implicated genes, for which we highlighted a contribution in the control of cancer cell features. Deepening into the Medulloblastoma biology, this study represents a further step forward for the development of molecular therapies in the era of precision oncology.

**Abstract:**

The impact of protein-coding genes on cancer onset and progression is a well-established paradigm in molecular oncology. Nevertheless, unveiling the contribution of the noncoding genes—including long noncoding RNAs (lncRNAs)—to tumorigenesis represents a great challenge for personalized medicine, since they (i) constitute the majority of the human genome, (ii) are essential and flexible regulators of gene expression and (iii) present all types of genomic alterations described for protein-coding genes. LncRNAs have been increasingly associated with cancer, their highly tissue- and cancer type-specific expression making them attractive candidates as both biomarkers and therapeutic targets. Medulloblastoma is one of the most common malignant pediatric brain tumors. Group 3 is the most aggressive subgroup, showing the highest rate of metastasis at diagnosis. Transcriptomics and reverse genetics approaches were combined to identify lncRNAs implicated in Group 3 Medulloblastoma biology. Here we present the first collection of lncRNAs dependent on the activity of the MYC oncogene, the major driver gene of Group 3 Medulloblastoma. We assessed the expression profile of selected lncRNAs in Group 3 primary tumors and functionally characterized these species. Overall, our data demonstrate the direct involvement of three lncRNAs in Medulloblastoma cancer cell phenotypes.

## 1. Introduction

Heterogeneity is a main feature of cancer and prevents the development of targeted therapies in the vast majority of cases. The recent molecular characterization of several tumors has greatly contributed to overcome this issue and has provided a new interpretation of each cancer type as a set of individual tumors. Unveiling the molecular basis underlying each tumor subtype is a requisite for instructing personalized cancer treatments. Such deeply molecular analysis has been successfully carried out in Medulloblastoma (MB), the most common malignant pediatric brain tumor [1] arising from progenitor cell populations during early brain development. It has been classified into the four molecular subgroups Wingless (WNT), Sonic Hedgehog (SHH), Group 3 (G3) and Group 4 (G4), with defined features both in terms of driver genes and clinical outcomes [2,3]. The most aggressive MB subgroup is the G3, accounting for 25% of all MBs and associated with the highest rate of metastasis at diagnosis (about 45%) and the worst survival outcome at 5 years (under 60%) [4,5]. Until now, a G3 common driver pathway has not been identified, which makes this subgroup still enigmatic. However, a c-MYC signature is restricted to G3: it occurs in about 17% of patients whereas it is extremely rare in other MB subgroups, G4 being characterized by n-MYC signature [2,3,6]. In particular, MYC alteration is due to both gene amplification and aberrant expression and is strictly related to unfavorable outcomes [4,7,8,9]. Notably, it has been recently demonstrated that the concomitant overexpression of *MYC* and *OTX*, another relevant driver gene in G3, is sufficient to induce MB in vivo [10].

c-MYC (referred to as MYC hereinafter) is an essential transcriptional factor that orchestrates the gene expression programs necessary for growth, expansion and homeostasis of somatic and stem cells [11]. Together with its protein partner MAX, and other cofactors, MYC drives transcription of at least 15% of all genes [12], both protein-coding and noncoding genes [13,14,15].

While in normal cells the levels of *MYC* RNA and protein are tightly regulated, in tumor cells they are aberrantly expressed. High levels of MYC have dramatic effects due to the amplification of the ongoing gene expression programs and the activation of previously silent genes implicated in cell cycle progression, proliferation, migration and metastasis [16]. For these reasons, MYC has long been considered an ideal cancer target [17]. Interfering with MYC expression or function was regarded as potentially detrimental for normal cells for a long time. However, recently the use of small peptides as MYC inhibitors proved to be a safe and effective therapeutic strategy [18]. Being deregulated in up to 70% of human cancers [19], MYC inhibition can be in principle used against multiple types of tumor.

By contrast, the development of tailored therapies implies a deeper knowledge of tumor-specific genes. The continued efforts in refining MB classification have provided a new challenge for identifying novel, specific tumor drivers beyond the well-known protein coding genes. In this framework, a specific class of noncoding transcripts, the long noncoding RNAs (lncRNAs), are emerging as attractive players in shaping MB features [20]. Because of their assessed role as oncogenes or oncosuppressors and their high tissue- and cancer-type specificity, they perfectly meet the criteria for being promising tumor targets and biomarkers. Nevertheless, for G3 MB mainly in silico studies have been carried out to pinpoint lncRNAs potentially involved in tumor biology [21]. To fill this gap, we aimed to underscore the lncRNAs that are regulated by MYC in a cell line derived from MYC-driven primary MB patient samples [22,23]. We inhibited MYC function, by the MYC dominant negative mutant OMOMYC [18,24], and carried out transcriptome analysis that produced the first atlas of MYC-dependent lncRNAs. Stringent filtering and experimental validation of the best candidates allowed us to select some transcripts whose involvement in cancer cell phenotypes was experimentally demonstrated.

## 2. Materials and Methods

### 2.1. Cell Culture

D283 MED cells (from ATCC) were cultured in MEM medium (M2279, Sigma-Aldrich, St. Louis, MO, USA) supplemented with 10% or 20% of heat-inactivated fetal bovine serum (USA origin), 1% of sodium pyruvate, 1% of non-essential amino acid solution, 1% of L-glutamine, and penicillin/streptomycin.

As for D283 MED cell propagation, floating cells were collected, whereas adherent ones were washed with PBS 1× w/o Ca^++^ and Mg^++^ and detached by incubation in 1× Trypsin/EDTA (T4299, Sigma-Aldrich) for 5 min. Cell population was centrifuged at 800 RPM for 5 min and the pellet was resuspended in fresh culture medium. Cells were counted on a CytoSMART Cell Counter (CLS6749, Corning, Sigma-Aldrich) and replated at the desired concentration.

### 2.2. Generation of D283-OMO Stable Cell Line

D283 MED cells harboring the doxycycline-inducible Flag-tagged OMOMYC transgene (D283-OMO) were obtained by lentiviral infection through the pSLIK-OMOMYC plasmid, described in Appendix A. Transduced cells were selected by 50 µL/mL Hygromycin B (10843555001, Sigma-Aldrich) and treated with 0.25 µg/mL doxycycline (D3447, Sigma-Aldrich) to induce OMOMYC expression.

### 2.3. Viable Cell Number Evaluation

D283 MED cells were seeded in six-well plates, treated according to experimental plans (OMOMYC induction or GapmeR transfection) and counted at different time points. Specifically, cells were thoroughly suspended and 10 µL aliquots were counted on a CytoSMART Cell Counter (CLS6749, Corning, Sigma-Aldrich) upon incubation with trypan blue dye.

### 2.4. LNA GapmeR Transfections

Control (LG00000002, Qiagen, Hilden, Germany) and specific LNA GapmeRs (339511, Qiagen) were transfected three times in DOX-untreated D283-OMO cells at 100 nM with Lipofectamine 2000 (11668-019, Invitrogen, ThermoFisher Scientific, Waltham, MA, USA), in opti-MEM I medium (31985070, Gibco, ThermoFisher Scientific), according to manufacturer’s instructions. Complete growing medium was added 5 h after transfection. LNA GapmeR sequences are listed in Appendix A.

### 2.5. RNA Extraction and Analysis

Total RNA was extracted by Direct-zol RNA MiniPrep (R2052, Zymo Research, Irvine, CA, USA). For quantitative real-time PCR (qRT-PCR) assay, cDNA was synthetized by Takara PrimeScript RT Reagent Kit (RR037A, Takara-bio, Kusatsu, Shiga Japan). qPCR detection was performed using SensiFAST SYBR Lo-ROX Kit (BIO-94020, Bioline, London, Great Britain) on a 7500 Fast Real-Time PCR (Applied Biosystem, Waltham, MA, USA). ATP5O was used as a reference target.

### 2.6. RNA-Seq and Bioinformatic Analysis

TruSeq Stranded mRNA Library Prep Kit (Illumina, San Diego, CA, USA) was used to obtain sequencing libraries from polyA + RNA extracted from D283-OMO and D283-OMO + DOX cells (4 indipendent biological replicates) through the miRNeasy Mini Kit (QG217004, Qiagen). The sequencing reaction, which produced 100 nucleotide long paired end reads, was performed on a Novaseq 6000 sequencing system (Illumina) with a depth of 80 M reads. RNA-Seq reads were trimmed using Trim Galore software (https://www.bioinformatics.babraham.ac.uk/projects/trim_galore/, accessed on 23 September 2019) to remove adapter sequences and low-quality end bases; the minimum read length after trimming was set to 20.

STAR [25] was employed to align reads to Genecode Human release 32 primary assembly. More than 90% of the reads were successfully mapped to the human genome most of them were aligned to unique locations (Appendix A).

The quantMode TranscriptomeSAM option was used to generate alignments translated into transcript coordinates. The expression levels of transcripts and genes were quantified by using RSEM [26], an accurate and user-friendly software tool for quantifying transcript abundances from RNA-Seq data. Differential expression analysis was performed using the Generalized Linear Model approach implemented in the Bioconductor package “edgeR” [27] on genes having more than 1 CPM in more than 4 samples. Genes with FDR < 0.001 and absolute logFC greater than 2 were considered differentially expressed and used for further analysis. The same bioinformatic pipeline was applied to analyze RNA-seq data from Luo et al. [28].

### 2.7. Chromatin Immunoprecipitation (ChIP) Assay

Chromatin extracts were prepared from 6 × 10^6^ DOX-untreated D283-OMO cells after crosslinking in 1% formaldehyde. Immunoprecipitation was performed using the MAGnify Chromatin Immunoprecipitation System kit (492024, Invitrogen, ThermoFisher Scientific), according to the manufacturer’s protocol. Briefly, chromatin extracts were sonicated and immunoprecipitated with 10 μL of rabbit anti-MYC antibody (9402, Cell Signaling Technology, Danvers, MA, USA), or rabbit IgG antibodies (provided by the kit) according to the manufacturer’s protocol (MAGnify Chromatin Immunoprecipitation System kit, 492024, Invitrogen, ThermoFisher Scientific).

Primer pairs for chromatin analysis were designed on lncRNA and BMP7 promoter regions combining MYC binding profiles from a ChIP-seq analysis carried out in MB03 cells [29] and prediction of genomic E-box sequences by JASPAR database (http://jaspar.genereg.net, ID: MA0147.3, accessed on 5 July 2021).

A standard curve was generated for each primer pair testing 5-point dilutions of input sample.

MYC and IgG enrichments on lncRNA promoters were quantified using qRT-PCR (PowerUp SYBR Green Master Mix, A25742, Life Technologies, Carlsbad, CA, USA) and calculated as a percentage of input chromatin.

### 2.8. Cell Fractionation

DOX-untreated D283-OMO cells were fractionated by the Ambion PARIS Kit (AM1921, Life Technologies). After RNA extraction, equal volumes of cytoplasmic or nuclear RNA were retro-transcribed and analyzed by qRT-PCR. Normalisations were based on the total amount of RNA.

### 2.9. Immunoblotting

Protein samples for immunoblotting were extracted from DOX-untreated D283-OMO cells in RIPA Buffer (50 mM Tris–HCl (pH 8), 150 mM EGTA, 150 mM NaCl, 50 mM NaF, 10% glycerol, 1.5 mM MgCl_2_, 1% Triton). Lysates were separated on gradient poly-acrylamide gels and transferred to Amersham Protran 0.45 um nitrocellulose membrane (10600002, GE Healthcare Life Sciences, Chicago, IL, USA), through the NuPAGE System (EI0002, Invitrogen). Immunoblots were incubated with the following antibodies: anti-GAPDH (sc-32233, Santa Cruz Biotechnology, Dallas, TX, USA); anti-FLAG (F1804, Sigma-Aldrich, S), anti-MYC (9402, Cell Signaling Technology, Danvers, MA, USA); anti-RHOT1 (sc-398520, Santa Cruz Biotechnology, Dallas, TX, USA); anti-cleaved PARP-1 (sc-56196, Santa Cruz Biotechnology); anti-cleaved CASPASE-3 (5A1E, Cell Signaling Technology).

Protein staining was performed by WesternBright ECL (K-12045-D50, Advansta, Menlo Park, CA, USA), detected by ChemiDoc XRS+ Molecular Imager (Bio-Rad, Hercules, CA, USA) and quantified through the Image Lab Software (release 3.0.1).

### 2.10. Flow Cytometry Analyses

To analyze cell-cycle phases distribution, both floating and adherent DOX-untreated D283-OMO cells were collected by centrifugation, fixed in cold 70% ethanol, and then stained in a PBS solution containing propidium iodide (PI; 62.5 μg/mL; P4864, Sigma-Aldrich), and RNase A (1.125 mg/mL; R6148, Sigma-Aldrich). Cell aggregates were gated out on bi-parametric graph FL-3lin/ratio as described [30]. Cell samples were analyzed in a Coulter Epics XL cytofluorometer (Beckman Coulter, Brea, CA, USA) equipped with EXPO 32 ADC software. At least 10,000 cells per sample were acquired. The percentage of cells in the different phases of cell-cycle and in sub-G1compartment was calculated using Flowing Software 2.5.1.

Apoptosis induction was analyzed by flow cytometry determination of Annexin V-FITC staining (556420, BD Biosciences, Franklin Lakes, NJ, USA)/PI (P4864, Sigma-Aldrich), to label necrotic or late apoptotic/dead cells with damaged cell membranes. Cell samples were analyzed as already described for cell cycle analysis.

### 2.11. BrdU Assay

DOX-untreated D283-OMO cells transfected with LNA GapmeRs were subjected to an 8 h pulse of BrdU. Cells were then fixed and stained for BrdU, according to the manufacturer’s protocol (11 296 736 001, Roche, Basel, Switzerland). Nuclei were counterstained with DAPI (28718-90-3, Sigma-Aldrich). Cells were captured with a microscope at 40× magnification and counted with ImageJ (version 1.53a). The number of BrdU-positive nuclei from three independent experiments was counted in 6 view-fields/replicate and the values averaged.

### 2.12. Migration and Invasion Assays

DOX-untreated D283-OMO cells transfected with LNA GapmeRs were starved in serum-free medium for 12 h. After starvation, 5 × 10^5^ cells were seeded in serum-free medium into the upper chamber of non-coated cell culture inserts (662638, ThinCert™ Cell Culture Inserts 24 Well, 8 μm pore size, Greiner Bio-One International, Kremsmünster, Austria). Medium supplemented with 20% serum was used as a chemo-attractant in the lower chamber. After 6 h incubation at 37 °C, cells that did not migrate through the pores were removed by a cotton swab. Cells on the lower surface of the membrane were fixed, stained with Crystal Violet (1159400025, Sigma-Aldrich) and captured with the Axioskop 2 plus microscope (Zeiss, made in Germany) at 10× magnification. Cell counting was performed with ImageJ. The number of migrating cells from three independent experiments was counted in 9 view-fields/membrane and the values averaged.

For invasion assay, DOX-untreated D283-OMO cells transfected with LNA GapmeRs were starved in serum-free medium for 6 h. After starvation, 5 × 10^5^ cells were seeded in serum-free medium into the upper chamber of cell culture inserts (662638, ThinCert™ Cell Culture Inserts 24 Well, 8 μm pore size, Greiner Bio-One International) coated with Matrigel (354277, Corning, NY, USA). Medium supplemented with 20% serum was used as a chemo-attractant in the lower chamber. After 20 h incubation at 37 °C, cells that did not migrate through the pores were removed by a cotton swab. Cells on the lower surface of the membrane were fixed, stained with Crystal Violet (1159400025, Sigma-Aldrich) and imaged with a Zeiss Axioscope microscope at 10× magnification. Afterwards, each chamber with the invaded cells was soaked into 10% acetic acid for 10 min to wash out the Crystal Violet. 100 μL/well 10% acetic acid was added into 96-well plates, and the absorbance was measured at a wavelength of 570 nm using a microplate reader (Glomax multi detection system, Promega, Madison, WI, USA).

### 2.13. Cell Pictures

Cell pictures were captured with ZOE Fluorescent Cell Imager (Bio-Rad, made in Singapore). The scale bar indicates 100 µm.

### 2.14. Statistical Analyses

Results are expressed as means ± SEM from biological replicates. Statistical differences were analyzed by two-tailed Student’s *t*-test. A *p*-value < 0.05 was considered statistically significant. * *p* < 0.05, ** *p* < 0.01, *** *p* < 0.001.

### 2.15. Oligonucleotides

Oligonucleotide sequences are listed in Appendix A.

## 3. Results

### 3.1. OMOMYC Is Able to Inhibit MYC Activity in D283 MED Cells

To uncover the lncRNAs potentially involved in G3 MB biology, we directed our efforts towards the identification of MYC-regulated transcripts. As a model system, we chose the long-established D283 MED cell line [31]. Derived from MYC-driven primary MB tumors, it is presently regarded as a cell system at the crossroad between G3 and G4 MB, reflecting in vitro the partial overlap observed between these tumor subgroups in vivo [32]. Importantly in the framework of this study, D283 MED cells turned out to be an easy-to-handle system, displaying an amplification of MYC levels (see Appendix A and Section 3.2, Appendix A and [32,33]). In this model system, we investigated how MB transcriptome responds to perturbation of MYC activity by exploiting OMOMYC, the largely characterized MYC dominant-negative mini-protein [18,24] (Appendix A).

To generate a doxycycline (DOX)-inducible cell line expressing OMOMYC, D283 MED cells were transduced with a recombinant lentivirus expressing OMOMYC (OMO) fused to a FLAG-tag [34] (Appendix A). This allowed us to verify DOX-dependent OMOMYC induction in D283-OMO cells by immunodetection. The Western blot analysis of Figure 1A (left panel) and its relative histogram (right panel) show a progressive and robust upregulation of OMOMYC up to 72 h upon DOX administration, compared to control conditions. As expected, according to OMOMYC mechanism-of-action [35], a dramatic decrease of MYC protein levels was also observed. Consistently with the reduction of MYC activity and expression, the RNA levels of two well-established and widespread activated MYC target genes, Carbamoylphosphate Synthetase 2 (CAD, Figure 1B, left panel) and Nucleolin (NCL, Figure 1B, right panel) were significantly downregulated by about 50% and 30%, respectively, upon 72 h-OMOMYC induction.

Finally, since *MYC* loss-of-function is documented to impair cell growth programs in many experimental systems [19], we verified the effect of MYC inhibition on D283 MED cell viability. As shown in Figure 1C, DOX-treatment reduced the number of viable D283-OMO cells (OMO+), carrying the OMOMYC transgene, but not of wild-type D283 MED cells (WT+). This effect was entirely ascribable to OMOMYC induction since, even at 72 h of treatment, the DOX-treated D283-WT line (WT+) and the untreated control (OMO−) displayed a comparable number of viable cells.

Altogether, these molecular and cellular analyses demonstrate that OMOMYC is able to inhibit MYC transcriptional activity and reduce cell viability in our model system.

### 3.2. The Inhibition of MYC Function Alters D283 MED Transcriptome

To ascertain the effect of MYC functional inhibition on the transcriptome of D283 MED cells, we profiled the expression of polyadenylated (polyA+) transcripts by RNA-Seq analysis, in untreated vs. D283-OMO cells treated with DOX for 72h.

We found 43,486 expressed polyA+ RNAs in D283-OMO cells (Count Per Million (CPM) > 1 in at least four samples), corresponding to 15,222 unique gene loci (Appendix A). Compared to control conditions, in MYC-inhibited cells we identified 1,084 differentially expressed genes (False Discovery Rate (FDR) < 0.01 and log_2_ Fold Change (FC) > |2|, Appendix A), as reported in the heatmap of Figure 2A. They included 205 putative long non-protein-coding genes (lncRNAs), classified by the abundance in distinct biotypes (Figure 2B) and equally distributed between upregulated and downregulated species (Figure 2C).

In order to select the best candidates for functional analyses, a number of filtering criteria were applied to the differentially expressed lncRNAs (see schematization in Figure 2D). We first considered the 50 top hits, based on their maximum level of expression (CPM) or expression variance (fold-change, FC) in DOX-treated D283-OMO cells vs. untreated cells. By crossing these 2 lists, we identified 25 (Appendix A) and 28 species (Appendix A) that were: (i) strongly downregulated (log_2_FC < −2.8) or upregulated (log_2_FC > 3.7) upon MYC inhibition, and (ii) highly expressed at least in one experimental condition (CPM > 164.9 or CPM > 128.7, respectively).

### 3.3. MYC-Dependent lncRNAs May Represent Potential Oncogenes or Oncosuppressors

The expression of the top 20 transcripts from Appendix A was analyzed for validation by qRT-PCR in untreated vs. DOX-treated (72 h) D283-OMO cells. This analysis was successful for 30 transcripts (Figure 3 and Figure 4). In parallel, to exclude any secondary effect due to doxycycline, their levels were also evaluated in untreated vs. DOX-treated D283-WT cells, which do not express OMOMYC. Notably, out of the 30 transcripts, 19 RNAs recapitulated the expression trends revealed by RNA-Seq analysis, highlighting an overall accuracy of 63%. We found that 14 out of the 19 validated targets were not affected by DOX-treatment (see WT+, black histograms in Figure 3 and Figure 4) and may, therefore, be assumed as MYC-regulated genes. Specifically, nine transcripts were downregulated upon MYC inhibition (Figure 3), whereas 5 were upregulated (Figure 4) in the same condition.

The expression levels of the 14 validated MYC-dependent lncRNAs were evaluated in a pool of 10 normal cerebella and compared to those assessed in untreated or DOX-treated D283-OMO cells. Among the 9 species downregulated by OMOMYC (Figure 5A, black histograms) 6 candidates—namely the antisense *AC116407.1*, the processed transcript *AC091182.1*, the pseudogenes *ACKR4P1* and *TRIM51FP*, the lncRNA *AP005901.5* and the lincRNA *AC010998.3*—displayed considerably higher levels in MYC-driven MB cells (OMO−) compared to healthy cerebella (CEREB) suggesting that they may represent putative oncogenes. The comparable lncRNA expression levels between healthy cerebella and MYC-inhibited D283 MED cells (OMO+) confirms that, at least in part, MYC is responsible for their deregulation in tumor cells.

Regarding the genes upregulated by OMOMYC, only the lncRNA *PAUPAR* (plotted with black columns in Figure 5B) displayed lower levels in MYC-driven MB cells (OMO−) compared to cerebella and reached expression levels comparable to cerebellum upon MYC inhibition (OMO+). These results support a potential role for *PAUPAR* as an oncosuppressor.

### 3.4. Selection of lncMB1, lncMB2 and lncMB3 for Functional Analyses

Among the 6 lncRNAs downregulated by OMOMYC at levels comparable to healthy human cerebella, we selected for further analyses 3 candidates, namely *AC116407.1*, *AC091182.1* and *AC010998.3* (highlighted in the heatmap of Appendix A). We based our choice on two parameters relevant for functional studies: (i) their genomic organization that is informative about their potential target genes (see Section 3.7), and (ii) the possibility of designing efficient antisense probes for loss-of-function experiments (see Section 3.5). For simplicity, we renamed them *lncMB1 (AC116407.1)*, *lncMB2 (AC091182.1*) and *lncMB3* (*AC010998.3*). An UCSC screen shot depicting their gene structure is reported in Appendix A.

To further characterize their dependence on MYC, a Chromatin immunoprecipitation (ChIP) assay was performed in DOX-untreated D283-OMO cells, a condition in which the three lncRNAs are highly expressed. As control, we used the deeply investigated bone morphogenetic protein-7 (BMP-7) gene as a direct target of MYC in D283 MED cells [36]. As shown in Figure 6A, the occupancy of MYC on the promoter regions of the three lncRNA genes was even higher compared to the BMP-7 gene promoter, indicating that they are direct targets of MYC.

The expression of *lncMB1*, *lncMB2* and *lncMB3* in primary MB tumors was then investigated by taking advantage of a recently published RNA-Seq dataset, containing transcriptomic profiles of 59 patient-derived primary tumors, representing all the MB subgroups, and 4 control samples [28]. Figure 6B shows that *lncMB2* was significantly upregulated in G3 compared to WNT, SHH and G4. Furthermore, its upregulation with respect to control supports its potential role as a G3 oncogene. A significant upregulation in G3 with respect to SHH and G4 was also observed for *lncMB1* (Figure 6C). Finally, even though statistically not significant, a trend of upregulation in G3 with respect to all the other samples was observed for *lncMB3* (Figure 6D). Consistently with these data, the upregulated expression of the three lncRNAs, already assessed in G3-derived D283 MED cells (Figure 5A), was also observed in other G3 cell lines, namely D341 and HD-MB03, compared to cerebellum (Appendix A).

### 3.5. Antisense Targeting of lncMB1, lncMB2 and lncMB3 Affects D283 MED Cell Survival

Subcellular localization is a parameter to be considered for inferring lncRNA biological functions. Therefore, the distribution of *lncMB1*, *lncMB2* and *lncMB3* was assessed in DOX-untreated D283-OMO cells by nucleus-cytoplasm fractionation and following qRT-PCR analysis (Figure 7). Compared to fractionation controls (Figure 7A), 85% of *lncMB1* transcript was localized in the cytoplasm (Figure 7B). Conversely, *lncMB2* and *lncMB3* were almost equally distributed between the nuclear and cytoplasmic compartments, the former being slightly more abundant in the nucleus, the latter in the cytoplasm (Figure 7B).

*LncMB1*, *lncMB2* and *lncMB3* functional involvement in G3 MB tumorigenesis was investigated in DOX-untreated D283-OMO cells through LNA GapmeR-mediated silencing [37,38]. Compared to transfections of scrambled LNA GapmeRs, used as negative control, the specific targeting of *lncMB1*, *lncMB2* and *lncMB3* significantly downregulated their expression by 40%, 70% and 60%, respectively (Figure 7C). In these conditions, we registered an alteration of cell morphology, i.e., a reduction of the number and size of the multicell aggregates typically generated by growing D283 MED cells (Appendix A). Interestingly, also the total number of both adherent and floating cell drastically decreased upon lncRNA depletion, as quantitatively assessed by evaluating the number of viable DOX-untreated D283-OMO cells at specific time points after LNA Gapmer delivery (Figure 8). We found that knockdown of *lncMB1* and *lncMB2* led to a reduction of viable cell numbers by about 50% and 40%, respectively, at day 5 (D5). More prominent was the outcome of *lncMB3* depletion, which caused a three-fold decrease of viable cell numbers at the same time point.

These data indicate that the three MYC-dependent lncRNAs affect G3 MB cell survival, with *lncMB3* displaying the strongest effect, which prompted further analyses to establish their involvement in tumor cell phenotypes.

### 3.6. LncMB3 Counteracts Apoptosis

To explore whether the effect of *lncMB1*, *lncMB2* and *lncMB3* knockdown on DOX-untreated D283-OMO cell viability could be ascribed to an alteration of cell proliferation, we analyzed this event by two distinct approaches. Initially, BrdU assay, which measures the rate of DNA replication, was performed upon lncRNA silencing. Visualization (Figure 9A, left panel) and measurement (Figure 9A, right panel) of the incorporated BrdU revealed that the BrdU/Dapi ratio was not modified upon downregulation of each lncRNA, indicating that they did not influence the efficiency of DNA duplication. Subsequently, cell cycle progression was thoroughly examined by propidium iodide (PI)-staining and flow cytometric analysis (Figure 9B, left panel). We found that *lncMB1*, *lncMB2* and *lncMB3* knockdown did not cause alterations in the distribution of cells among G0-G1, S and G2-M cell cycle phases with respect to the control condition (Figure 9B, right panel). All together, these data indicate that the proliferation rate of D283 MED cells was not regulated by *lncMB1*, *lncMB2* and *lncMB3*.

We next explored a possible involvement of *lncMB1*, *lncMB2* and *lncMB3* in programmed cell death (apoptosis), by evaluating the number of cells accumulated in the sub-G1 peak [39]. No significant variation in DNA fragmentation in the sub-G1 peak was observed in DOX-untreated D283 MED cells upon *lncMB1* and *lncMB2* downregulation (Figure 9C). Conversely, cell accumulation in the sub-G1 peak increased by about 40% upon *lncMB3* depletion, suggesting its anti-apoptotic activity in D283 MED cells.

The role of *lncMB3* in apoptosis was verified more in detail by PI- and Annexin V-staining of DOX-untreated D283-OMO cells, following lncRNA depletion. When incorporated in DNA, PI reveals the rupture of cell membranes occurring in late apoptosis or necrosis, whereas Annexin V selectively marks apoptotic cell surface. The distribution of cells in different conditions (early or late apoptotic, necrotic, or viable) was evaluated by flow cytometric determination, as reported in the diagrams of Figure 10. The corresponding cell percentages were quantified in the histogram of Figure 10A, right panel, which shows: (i) a significant decrease of the fraction of viable cells, negative to both Annexin and PI, upon *lncMB3* knockdown and (ii) a parallel, strong increase of both early and late apoptotic cells (positive to Annexin V and negative or positive to PI, respectively). In line with previous results, no significant variations were registered upon silencing of *lncMB1* and *lncMB2*.

To further corroborate our results, the levels of cleaved Poly (ADP-ribose) polymerase 1 (cPARP-1) protein, a well-characterized product of caspase activity [40,41], were analyzed by Western blot upon lncRNA depletion. Compared to control cells, an increase by about ten-folds of cPARP-1 level was obtained upon *lncMB3* targeting (Figure 10B); no significant variation was detected upon *lncMB1* and *lncMB2* silencing, suggesting that *lncMB3* depletion specifically triggers caspase-dependent apoptosis. This result was strengthened by the analysis of Caspase-3, that is mainly responsible for PARP cleavage during the apoptotic process. Indeed, the active subunit of Caspase-3 (cCas-3) was increased by 3-fold compared to the control upon *lncMB3* knockdown (Figure 10C), thus confirming that *lncMB3* depletion triggers a caspase-dependent apoptosis in MB cells [42].

### 3.7. LncMB2 and lncMB1 Control ZNF703 and RHOT1 Gene Expression, Respectively

Next, we addressed the function of *lncMB2* and *lncMB1*. As stated before, *lncMB2* is a predominantly nuclear transcript, whose proximal genes *ZNF703* and *ADGRA2*, mapping about 360 Kb and 460 Kb apart from *lncMB2 locus (AC091182.1)*, respectively (Appendix A), are both related to cancer [43,44]. qRT-PCR analysis of their expression highlighted a down-regulation by about 30% of *ZNF703* RNA levels upon *lncMB2* silencing, compared to scramble-transfected D283 MED cells (Figure 11A, left panel). In the same conditions, we found a 50% reduction of *ADGRA2* expression (Figure 11A, right panel). These results indicate that *lncMB2* acts as a positive regulator of the oncogene *ZNF703* and the CNS angiogenesis regulator *ADGRA2*. Since ZNF703 activity as a transcriptional corepressor is associated with cell migration and adhesion [43] and given that G3 MB shows the highest metastasis rate among MB subgroups [5,7,45,46], we measured migration and invasion capacity of DOX-untreated D283-OMO cells by a trans-well assay. Visualization and count of crystal violet-stained cells highlighted a significant decrement of both migration (Figure 11B) and invasion (Figure 11C) by about 30% upon *lncMB2* knockdown with respect to control. Our data prove that *lncMB2* favors MB cell mobility, suggesting a possible role in promoting metastatic behaviors.

With regard to *lncMB1*, as mentioned before, it is a cytoplasmic transcript, antisense to *Rhot1* gene (Appendix A), encoding for a RAS protein involved in mitochondrial homeostasis, apoptosis and cancer [47,48,49,50].

We hypothesized that *lncMB1* may regulate stability and/or translation of its sense transcript. qRT-PCR analysis showed no significant modulation of *RHOT1* mRNA levels upon *lncMB1* knockdown compared to control (Figure 12A). However, we observed a 30%-decrease of RHOT1 protein (Figure 12B) in the same conditions. This suggests a role for *lncMB1* as a positive regulator of *RHOT1* gene at the translational and/or protein stability level.

## 4. Discussion

Cancer is a very complex pathology characterized by high heterogeneity among patients and tumor types [51,52]. It is well established that both coding and non-coding mutations greatly contribute to cancer biology. In particular, the vast majority of known driver mutations affect protein-coding regions and are mainly responsible for aberrant chromatin remodeling and proliferation pathway alterations [53]. Nevertheless, a number of genetic perturbations that affect developmental pathways, such as WNT and NOTCH, are also produced by somatic mutations in non-coding regions as the *cis* regulatory (promoters) and enhancer sequences or the untranslated regions (5′ and 3′UTRs), all of which may have a strong impact on gene expression [53].

Moreover, recent studies that combined DNA- and RNA-based approaches to identify cancer-associated pathways have greatly expanded our knowledge of the multiple mechanisms underlying tumor biology. They revealed that some alterations may occur through changes in RNA, rather than DNA sequence mutations, such as overexpression [54], altered splicing [55] and gene fusion [56].

To add a further layer of complexity, genomic alterations in coding genes that control multiple targets—as chromatin regulators or transcription factors—may in turn affect noncoding genes. This is the case of lncRNAs, noncoding transcripts that are widely implicated in the regulation of gene expression programs underlying relevant biological processes, such as cell differentiation and development. They may act both in *cis* and in *trans* and, because of their modular nature, have the unique property to interact with proteins as well as with nucleic acids, both DNA and RNA, with high specificity. Notably, their ability to simultaneously establish such interactions provides them the possibility of targeting specific factors/complexes to a single location [57,58].

In line with their roles, the aberrant expression of lncRNAs may profoundly affect cellular pathways with pathological outcomes. Nowadays, they have been increasingly implicated in tumorigenesis and have been shown to contribute to each of the cancer hallmarks, from cell proliferation and survival to apoptosis, invasion and angiogenesis [59,60]. Such implication arose from the observation that lncRNA expression may be regulated by key oncogenic transcription factors such as MYC, which is involved in the majority of human tumors [19,61]. This suggested the possibility that these transcripts may play a part in the functional output of the oncogenic signal. Notably, while MYC is expressed in a variety of tumors, the lncRNAs are endowed with cell- and cancer type-specific expression which suggests that the same oncogene MYC may influence the expression of distinct sets of lncRNAs, depending on the pathological context. Therefore, while MYC is regarded as a universal target in cancer [18], the identification of MYC-responsive lncRNAs in distinct tumors may represent an alternative strategy for unveiling novel biomarkers as well as driver genes and therapeutic targets. They may be powerful biomarkers not only because many of them are uniquely expressed in distinct cancer types [62], but also because they may be easily detected in body fluids, such as urine, blood and cerebrospinal fluids, making the tumor diagnosis less invasive [63]. Their activity as driver genes is tightly dependent on their role as crucial nodes of regulatory networks. Paradigmatic is their action as microRNA sponges that may derepress gene expression in a pleiotropic manner [64,65] or as scaffolds to deliver transcriptional factors or chromatin remodeling complexes to the chromatin site [66,67].

In the era of precision oncology, we decided to unveil the contribution of lncRNAs to MB, the most common pediatric malignant brain tumor, mainly occurring in children under the age of ten [68]. To date, no pharmacological approaches are decisive in the treatment of this tumor, while the secondary effects of chemotherapy, craniospinal radiation or surgical interventions heavily affect the quality of life of pediatric patients, requiring the rapid development of alternative therapies. In particular, G3 MB subgroup sparked our interest for a number of reasons: (i) it is the most aggressive subgroup, being associated with the highest rate of metastasis at diagnosis (40–45%) and the worst survival outcome (under 60% at 5 years) [4,5]; (ii) no any univocal driver pathway has been underscored so far; (iii) the lncRNA landscape has never been thoroughly explored [69].

To unveil the lncRNAs engaged in G3 MB tumorigenesis, we exploited a feature of this subgroup, namely the high MYC level, due to both gene copy number increase and aberrant expression [3,10]. Therefore, we looked for MYC-dependent lncRNAs in a cell line, the D283 MED cells, in which *MYC* is overexpressed [32].

To this aim, we inhibited MYC function through the well-characterized dominant-negative OMOMYC [18,24] and analyzed the resultant impact on cell transcriptome. The advantage of using OMOMYC strategy is at least double. On the one hand, by sequestering MYC away from E-boxes on the promoter regions of target genes, OMOMYC blocks the expression of the MYC gene signature, common to tumors with high MYC expression [70]. Additionally, OMOMYC was also proposed to form transcriptionally inactive homodimers to E-boxes, resulting in inhibition of MYC target gene expression [18]. On the other hand, OMOMYC, interfering with the binding of MYC to its partner MAX, leads to ubiquitination and proteasome-dependent degradation of the free MYC monomer [35].

This strategy allowed us to compile the first atlas of MYC-dependent lncRNAs in G3 MB [10]. Through a stringent filtering procedure, we selected three candidates, renamed *lncMB1*, *lncMB2* and *lncMB3*, to be tested for their involvement in G3 MB biology.

By comparing their expression profile in MYC-driven MB-derived cells and G3 primary tumors to normal cerebella, we hypothesized a potential oncogenic role for all of them.

By testing the ability of the lncRNAs to influence tumor cell-related features, we highlighted a role for *lncMB3* in evading programmed cell death. Apoptosis, as a protective mechanism devoted to the maintenance of tissue homeostasis, represents a natural barrier that should be circumvented during tumor development [71]. Accordingly, the acquired resistance towards apoptosis is a hallmark of most, if not all, types of cancer. In this context, the future discovery of *lncMB3* target genes will provide a new molecular pathway underlying G3 MB pathogenesis.

Starting from the discovery of the oncogene ZNF703 [43,72,73,74] as a *lncMB2* target gene, we were able to demonstrate a role for the lncRNA in promoting cell migration and invasion, processes underlying cancer metastasis [75]. However, further studies are needed to realize whether this capability is mediated by ZNF703 and/or by other regulatory circuitries.

We demonstrated for *lncMB1* a regulative role in the expression of its sense gene *RHOT1.* RHOT1 is implicated in an MYC-regulated gene network underlying mitochondrial trafficking, whose alteration has been recently considered as a hallmark of MYC-driven tumors [76]. In particular, it was demonstrated that the accumulation of mitochondria at the cortical cytoskeleton of tumor cells promotes cell invasion and metastasis [76]. It should be speculated that *lncMB1* might be a novel component of this network in which MYC may control the essential component *RHOT1* both directly and indirectly through the lncRNA.

## 5. Conclusions

This study uncovered remarkable activities for the three MYC-dependent lncRNAs, *lncMB1*, *lncMB2* and *lncMB3*, in a G3 MB landscape, and identified their target gene(s), which will further future studies aimed at building novel regulative circuits that are altered in this aggressive pediatric tumor.

## Figures and Tables

**Figure 1 cancers-13-03853-f001:**
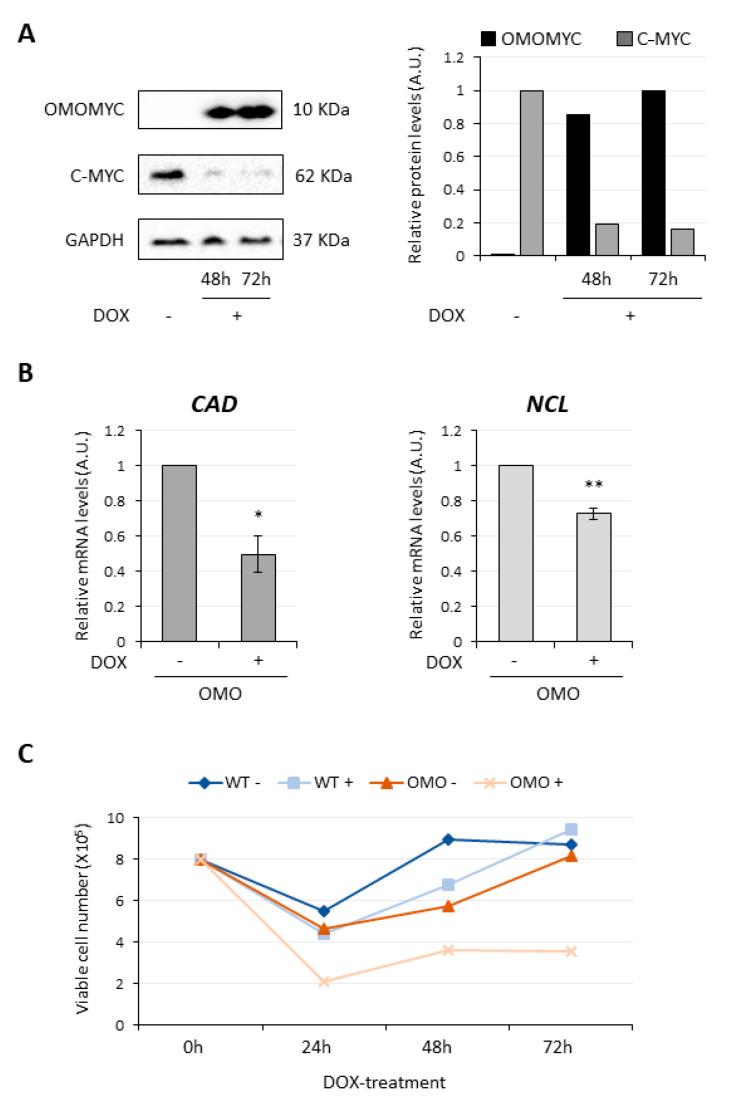
OMOMYC inhibits MYC expression and function in D283 MED cells. Details in Appendix A. (**A**) Left panel: immunoblot analysis of OMOMYC and MYC proteins in DOX-untreated (lane −) or treated (lanes +) D283–OMO cells; GAPDH was used as control. Right panel: quantification of OMOMYC (black bars) and MYC (grey bars) protein levels expressed in arbitrary units (A.U.) relative to GAPDH. *N = 1*. Timing of DOX treatment (hours) is reported below each lane and each corresponding histogram bar. (**B**) qRT-PCR analysis of *CAD* (left panel) and *NCL* (right panel) gene expression upon 72h-OMOMYC induction by DOX-treatment (+) in D283-OMO cells. Untreated cells (−) were used as control and set as 1. Data (means ± SEM) are expressed in arbitrary units (A.U.) and are relative to *ATP5O* mRNA levels. *N* = 3, * *p* ≤ 0.05, ** *p* ≤ 0.01 (two-tailed Student’s *t*-test). *(***C**) Analysis of the number of viable cells in D283-OMO (OMO) and D283-WT lines (WT), untreated (−) or treated (+) with DOX for 0, 24, 48 and 72 h. Viable cells were counted using an automated cell counter. *N* = 1.

**Figure 2 cancers-13-03853-f002:**
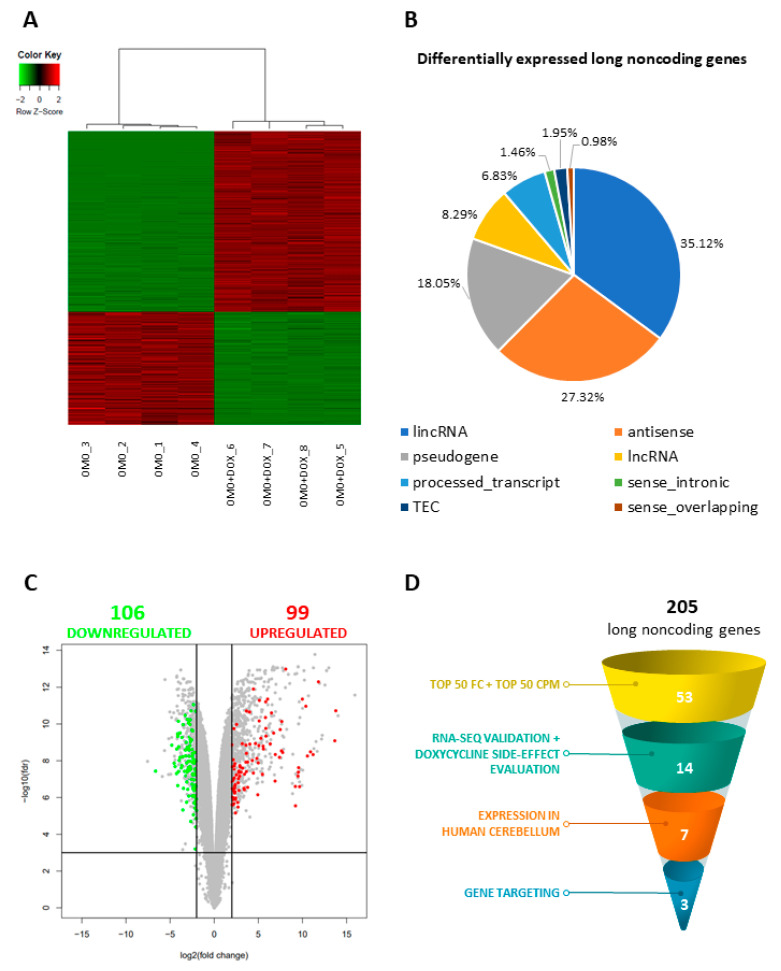
OMOMYC induction affects the transcriptome of D283 MED cells. (**A**) Heatmap shows the relative levels of differentially expressed genes in untreated (OMO) vs. DOX-treated (72 h) D283-OMO cells (OMO + DOX), along with sample (four biological replicates for each condition) hierarchical clustering. Expression levels in the heatmap were calculated by mean-centering the CPM values for each gene. The heatmap represents only the genes displaying CPM > 1 in at least four samples, FDR < 0.01 and log_2_FC > |2|. (**B**) Biotype abundance of the long non-coding genes identified as differentially expressed in untreated vs. DOX-treated D283-OMO cells according to RNA-Seq data analysis (CPM > 1 in at least four samples, FDR < 0.01 and log2FC > |2|). (**C**) Volcano plot showing the distribution of the genes differentially expressed in untreated vs. DOX-treated D283-OMO cells, according to RNA-seq data analysis. Genes were plotted on the basis of statistical significance [−log_10_(FDR)] and expression variance [log_2_FC]. Non-coding genes (205 hits, CPM > 1 in at least four samples, FDR < 0.01 and log_2_FC > |2|) are highlighted by green (downregulated species) or red (upregulated species) dots, respectively. Protein-coding genes are indicated in grey. (**D**) Schematic representation of the filtering criteria applied to MYC-dependent lncRNAs to select candidates for functional analyses.

**Figure 3 cancers-13-03853-f003:**
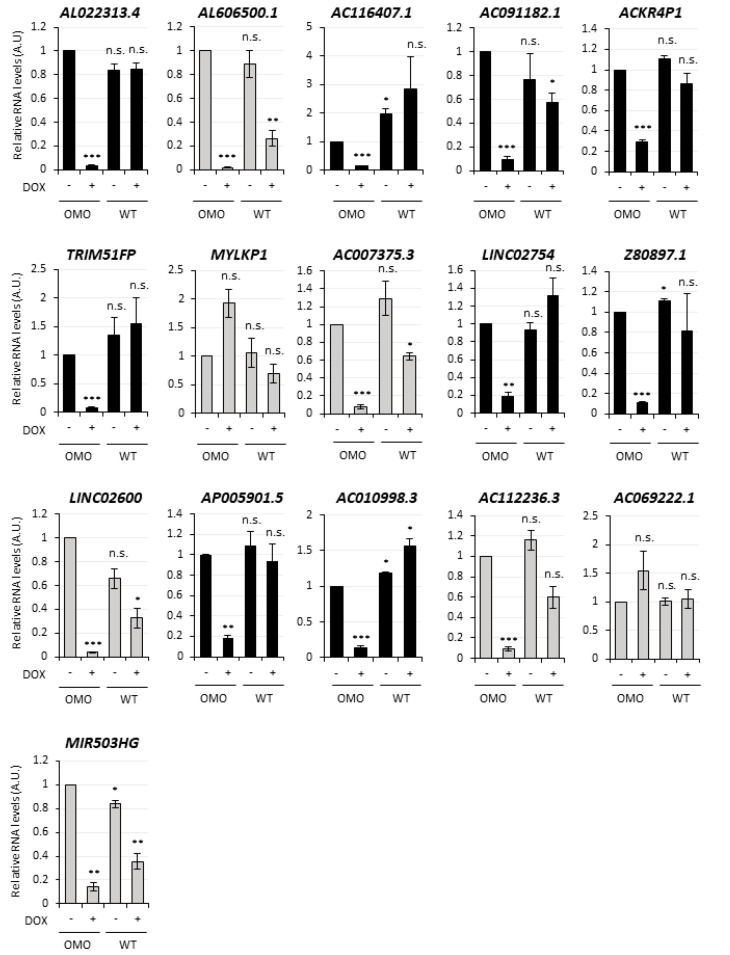
Validation of the MYC-dependent lncRNAs downregulated by OMOMYC in D283 MED cells. qRT-PCR analysis in D283-OMO cells 72 h-treated with DOX (OMO+) of the most expressed and downregulated lncRNAs, according to the RNA-Seq. Among the 20 top hits from the Appendix A 16 RNAs were successfully detected included in the figure. Expression levels were compared to untreated D283-OMO (OMO−) as control, set as 1. Parallel analysis in untreated (WT−) vs. treated WT (WT+) D283 MED cells revealed possible unspecific effects of DOX on lncRNA expression. Validated candidates (9 lncRNAs) are indicated by black histograms, non-validated candidates (7 lncRNAs) are reported in grey. Data (means ± SEM) are expressed in arbitrary units and are relative to *ATP5O* mRNA levels. *N* = 3, * *p* ≤ 0.05, ** *p* ≤ 0.01, *** *p* ≤ 0.001 (two- tailed Student’s *t*-test).

**Figure 4 cancers-13-03853-f004:**
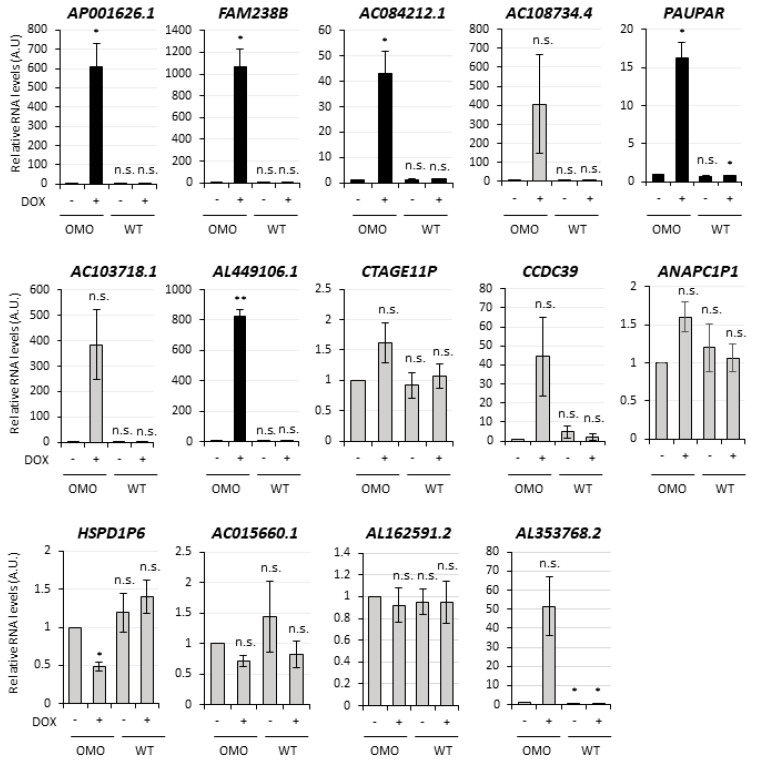
Validation of the MYC-dependent lncRNAs upregulated by OMOMYC in D283 MED cells. qRT-PCR analysis in D283-OMO cells 72 h treated with DOX (OMO+) of the most expressed and upregulated lncRNAs, according to the RNA-Seq (see Appendix A). Among the 20 top hits from the table, 14 RNAs were successfully detected and included in the figure. Expression levels were compared to untreated D283-OMO (OMO−) as control, set as 1. Validated candidates (5 lncRNAs) are indicated by black histograms, non-validated candidates (9 lncRNAs) are reported in grey. Data (means ± SEM) are expressed in arbitrary units and are relative to *ATP5O* mRNA levels. *N* = 3, * *p* ≤ 0.05, ** *p* ≤ 0.01 (two- tailed Student’s *t*-test).

**Figure 5 cancers-13-03853-f005:**
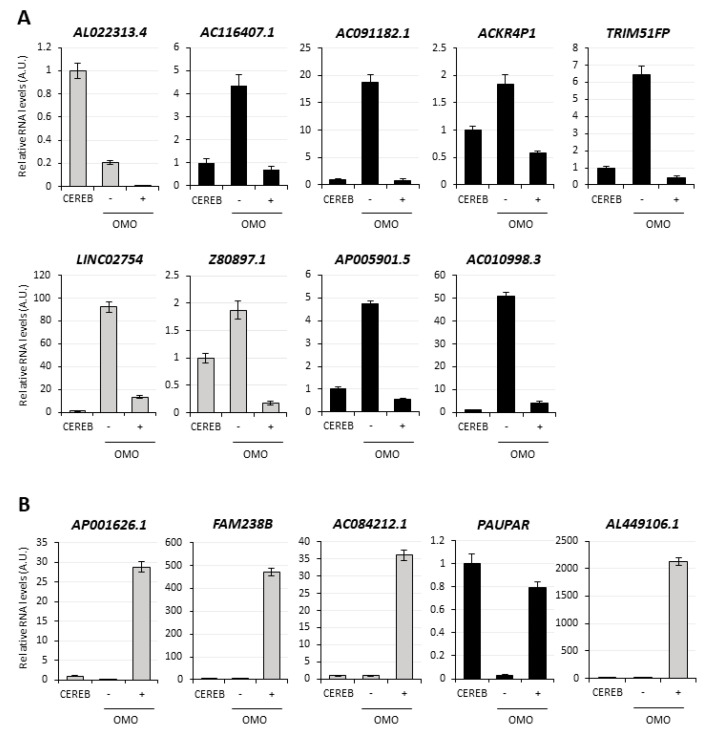
MYC-regulated lncRNA profiling in healthy cerebellum RNA samples. (**A**) Comparison of the expression profile, by qRT-PCR, of the 9 lncRNAs downregulated in D283-OMO cells 72 h-treated with DOX (OMO+) compared to healthy cerebellum RNA samples (CEREB set as 1). As control, the analysis in untreated D283-OMO cells (OMO−) was also carried out. Data (means ± SEM from technical replicates) are expressed in arbitrary units (A.U.) and are relative to *ATP5O* mRNA levels. *N* = 1. (**B**) qRT-PCR analysis in healthy cerebellum RNA samples (CEREB, set as 1) of the 5 lncRNAs upregulated in D283-OMO cells upon OMOMYC induction (see Figure 4). Details as in panel (**A**).

**Figure 6 cancers-13-03853-f006:**
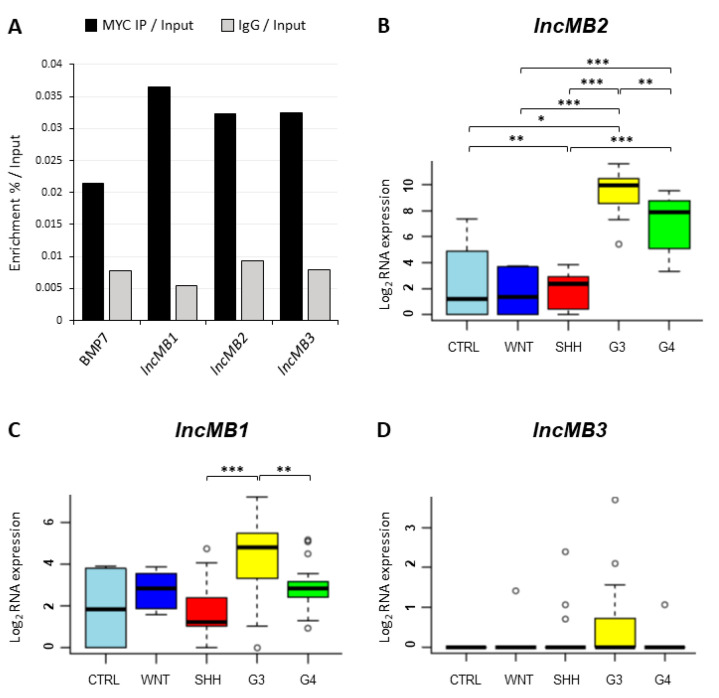
MYC-regulated *lncMB1*, *lncMB2* and *lncMB3* expression in primary MB tumors. (**A**) MYC occupancy on regions upstream to *lncMB1*, *lncMB2* and *lncMB3* transcriptional start site in DOX-untreated D283-OMO cells. BMP7 promoter region was used as positive control. IP and IgG enrichments are expressed as percentage relative to input. N = 1. (**B**–**D**) Expression levels of *lncMB2* (panel **B**), *lncMB1* (panel **C**) and *lncMB3* (panel **D**) in 4 control (CTRL) samples and 59 primary MBs, assessed from RNA-Seq data deposited by Luo et al., 2021 [28]. Results are expressed in log_2_(FPKM). * *p* ≤ 0.05, ** *p* ≤ 0.01, *** *p* ≤ 0.001 (two-tailed Student’s *t*-test).

**Figure 7 cancers-13-03853-f007:**
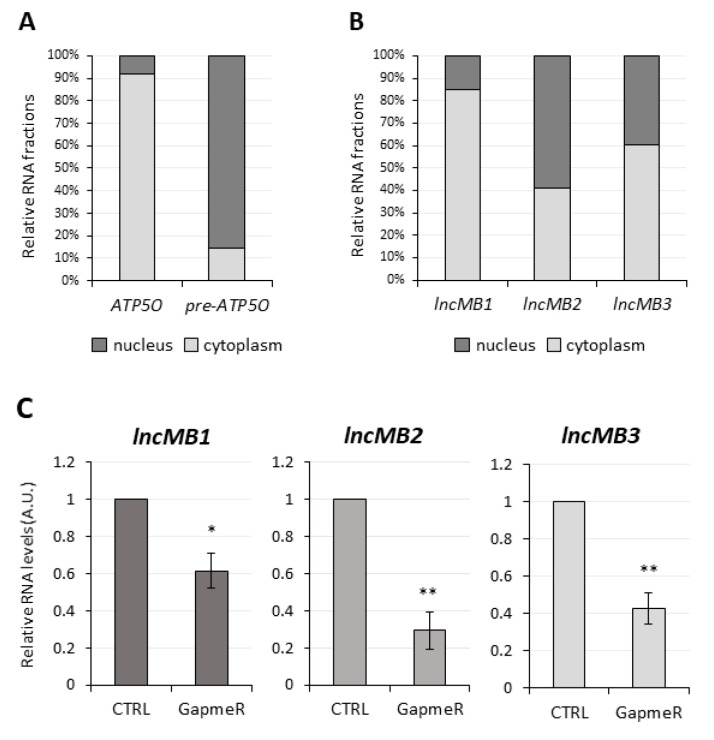
Subcellular localization and knock-down of *lncMB1*, *lncMB2* and *lncMB3* in DOX-untreated D283-OMO cells. (**A**) qRT-PCR analysis of *ATP5O* and *pre-ATP5O* transcripts in nuclear (dark grey bars) and cytoplasmic (light grey bars) fractions from DOX-untreated D283-OMO cells. Normalizations were performed on the total amount of RNA. Data are expressed as a percentage of the total levels of *ATP5O* or *pre-ATP5O*. *N* = 1. (**B**) qRT-PCR analyses of *lncMB1*, *lncMB2* and *lncMB3* expression in nuclear (dark grey bars) and cytoplasmic (light grey bars) fractions derived from DOX-untreated D283-OMO cells. Details as in (**A**). (**C**) qRT-PCR analyses of *lncMB1* (**left** panel), *lncMB2* (**middle** panel), and *lncMB3* (**right** panel), expression levels upon specific LNA GapmeR transfections in DOX-untreated D283-OMO cells. Expression was set as 1 in scrambled-transfected control cells (CTRL). Data (means ± SEM) are expressed in arbitrary units (A.U.) and are relative to *Atp5o* mRNA. *N* = 4, * *p* ≤ 0.05, ** *p* ≤ 0.01 (two-tailed Student’s *t*-test).

**Figure 8 cancers-13-03853-f008:**
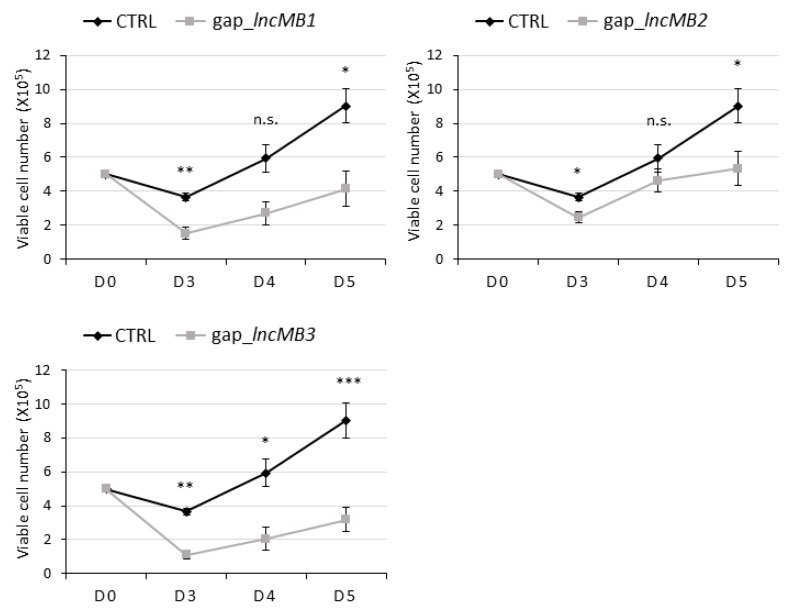
*LncMB1*, *lncMB2* or *lncMB3* knock-down affects viable DOX-untreated D283-OMO cell number. Number of viable DOX-untreated D283-OMO cells depleted for *lncMB1* (upper left panel), *lncMB2* (upper right panel), and *lncMB3* (lower left panel) was measured in a time course experiment. Scramble-transfected cells (CTRL) were used as control. Cell counts started after the last LNA-GapmeR transfection pulse (i.e., day 3 after cell seeding). Data (means ± SEM) are expressed as the number of viable cells, counted by an automated cell counter. *N* = 4, * *p* ≤ 0.05, ** *p* ≤ 0.01, *** *p* ≤ 0.001 (two-tailed Student’s *t*-test). Cell morphology is reported in Appendix A.

**Figure 9 cancers-13-03853-f009:**
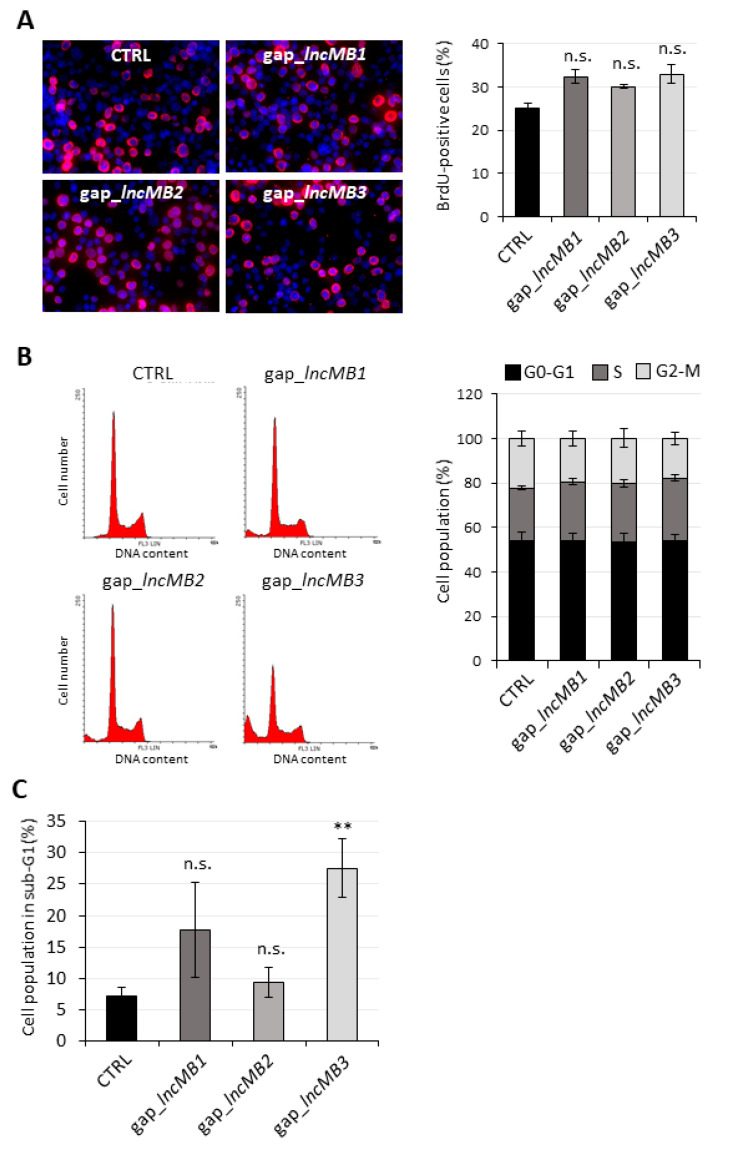
DOX-untreated D283-OMO cell proliferation and cell cycle are not affected by *lncMB1*, *lncMB2* and *lncMB3* knock-down. (**A**) Left panel: representative pictures of DOX-untreated D283-OMO cells knocked down for *lncMB1*, *lncMB2* and *lncMB3* and subjected to a BrdU incorporation assay, compared to scramble-transfected cells. Magnification 10×. BrdU-positive cells are visualized in red, nuclei are counterstained in blue, merge signals appear as pink. Right panel: quantification of BrdU-incorporating DOX-untreated D283-OMO cells knocked down for *lncMB1*, *lncMB2* or *lncMB3*, compared to scramble-transfected (CTRL) cells. Data (means ± SEM) from 6 view-field/replicate are expressed as a percentage over the total number of cells. BrdU-positive cells and nuclei were counted with ImageJ. *N* = 3. (**B**) Left panel: representative cell cycle analysis by flow cytometric determination of PI-stained DOX-untreated D283-OMO cells upon depletion of *lncMB1*, *lncMB2* and *lncMB3*, compared to control cells. DNA content and cell number are reported on *x*- and *y*-axes of the diagrams, respectively. Right panel: quantification of the of PI-stained DOX-untreated D283-OMO cells (knocked down for *lncMB1*, *lncMB2* or *lncMB3*) according to their distribution in G0/G1, S and G2-M cell cycle phases, established by flow cytometry. Scramble-transfected (CTRL) cells were used as control. Data (means ± SEM) are expressed as percentages over the total cell number. *N* = 4. Statistical test (two-tailed Student’s *t*-test) was performed by comparing, for each phase of the cell cycle, cell percentages between experimental vs. control conditions. (**C**) Quantification of cells in the sub-G1 phase (data from Figure 4, left panel). Data (means ± SEM) are expressed as percentages over the total cell number. *N* = 4, ** *p* ≤ 0.01 (two-tailed Student’s *t*-test).

**Figure 10 cancers-13-03853-f010:**
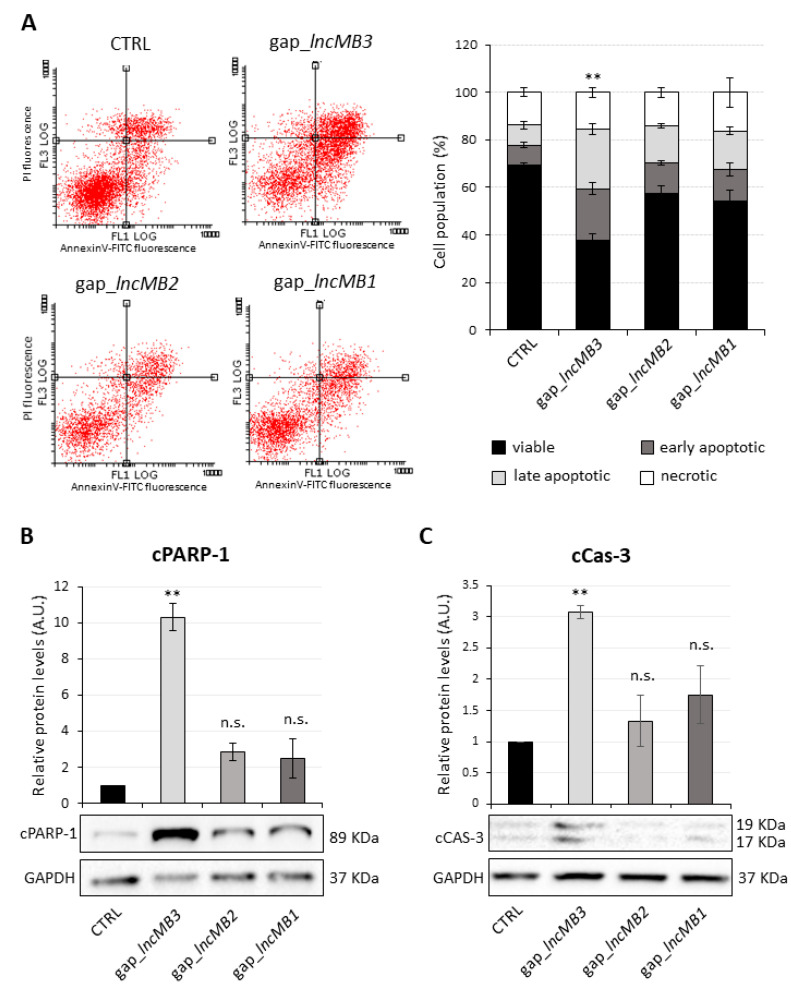
*LncMB3* counteracts apoptosis in DOX-untreated D283 MED cells. (**A**) Left panel: representative flow cytometry analysis of PI- and Annexin- stained DOX-untreated D283-OMO cells upon *lncMB3*, *lncMB2* and *lncMB1* knock-down. PI and AnnexinV-FITC fluorescence are reported on *x*- and *y*-axes of the diagrams, respectively. Right panel: quantification of the fractions of viable (black bars), early apoptotic (dark grey bars), late apoptotic (light grey bars) and necrotic (white bars) cells. Data (means ± SEM) are expressed as percentages over the total cell number. *N* = 4, ** *p* ≤ 0.01 (two-tailed Student’s *t*-test). Statistical test was performed by comparing, for each cell subpopulation, cell percentage in experimental vs. control conditions. (**B**) Representative immunoblot analysis of cPARP-1 protein levels upon *lncMB3* knock-down in DOX-untreated D283-OMO cells, compared to scramble-transfected cells. Cells depleted for *lncMB1* and *lncMB2* were used as specificity controls. In the histogram above, cPARP-1 levels are quantified relative to GAPDH. Scramble-transfected cells were set as 1. Data (means ± SEM) are expressed in arbitrary units (A.U.). *N* = 3, ** *p* ≤ 0.01 (two-tailed Student’s *t*-test). (**C**). Representative immunoblot analysis of cleaved Caspase-3 (cCas-3) protein levels upon *lncMB1*, *lncMB2* and *lncMB3* knock-down in DOX-untreated D283-OMO cells, compared to scramble-transfected cells. Details as in panel (**B**).

**Figure 11 cancers-13-03853-f011:**
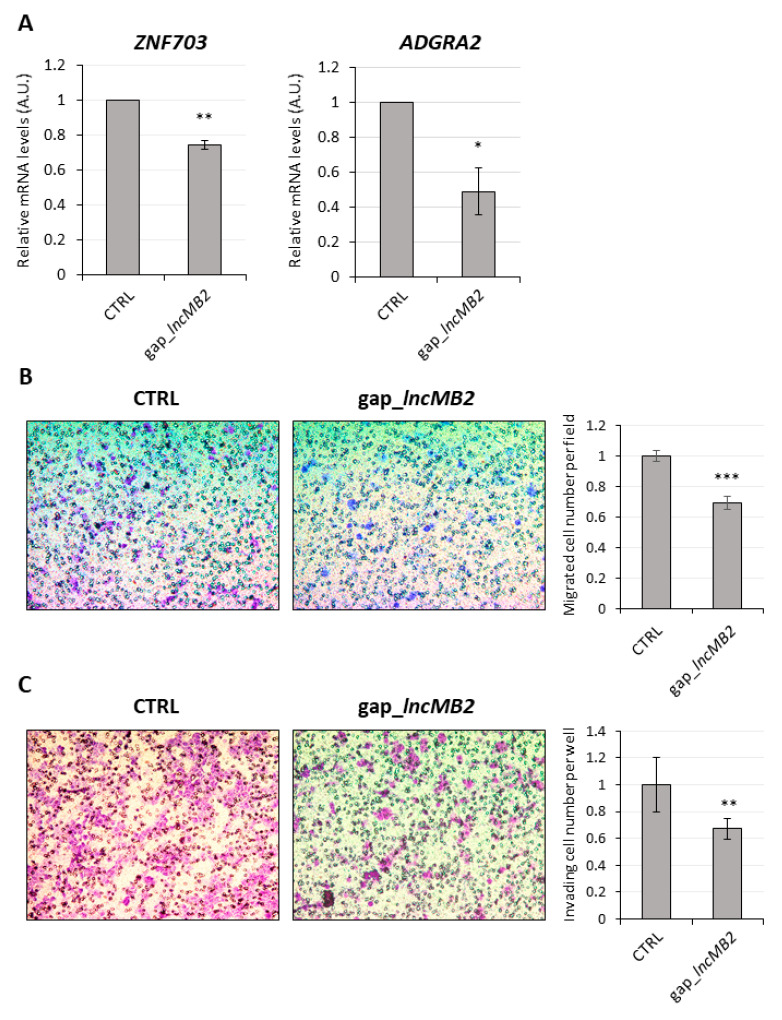
*LncMB2* activates *ZNF703* and *ADGRA2* gene expression and promotes migration/invasion of DOX-untreated D283-OMO cells. (**A**) qRT-PCR analysis of *ZNF703* (left panel) and *ADGRA2* (right panel) expression in DOX-untreated D283-OMO cells upon knock down of *lncMB2*, compared to control transfected cells (CTRL), set as 1. Data (means ± SEM) are expressed in arbitrary units (A.U.) and are relative to *ATP5O* mRNA. *N* = 4, * *p* ≤ 0.05, ** *p* ≤ 0.01 (two-tailed Student’s *t*-test). (**B**) Left panel: brightfield visualization of crystal violet-stained DOX-untreated D283-OMO migrating cells upon depletion of *lncMB2*. Scramble LNA GapmeR-transfected cells were used as a control. Magnification 10×. Right panel: cells were counted by ImageJ in 9 view-fields/membrane and quantified in the histogram. Data (means ± SEM) are expressed as number of migrated cells. *N* = 3, *** *p* ≤ 0.001 (two-tailed Student’s *t*-test). (**C**) Left panel: brightfield visualization of crystal violet-stained DOX-untreated D283-OMO invading cells upon *lncMB2* depletion. Details as in B. Magnification 10×. Right panel: quantification of invading cell number was determined by colorimetric assay. *N* = 3, ** *p* ≤ 0.01 (two-tailed Student’s *t*-test).

**Figure 12 cancers-13-03853-f012:**
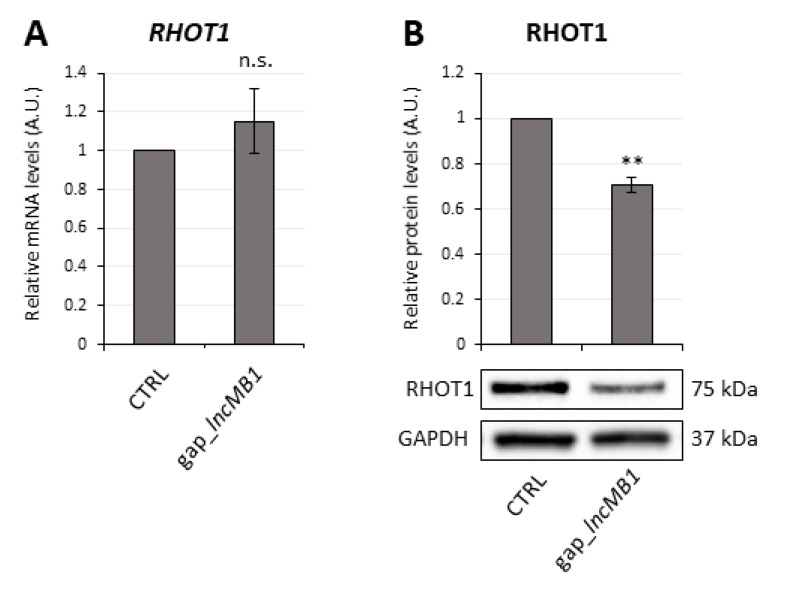
*LncMB1* controls RHOT1 protein levels in DOX-untreated D283 MED cells. (**A**) qRT-PCR analysis of *RHOT1* expression in DOX-untreated D283-OMO cells upon depletion of *lncMB1*, compared to scramble-transfected cells, set as 1. Data (means ± SEM) are expressed in arbitrary units (A.U.) and are relative to *ATP5O* mRNA. *N* = 4 (two-tailed Student’s *t*-test). (**B**) Representative immunoblot analysis of RHOT1 protein levels in DOX-untreated D283-OMO cells upon *lncMB1* knock-down, compared to scramble-transfected cells. In the histogram above, RHOT1 protein levels were quantified, relative to GAPDH. RHOT1 level in scramble-transfected cells, used as control, was set as 1. Data (means ± SEM) are expressed in arbitrary units (A.U.). *N* = 4, ** *p* ≤ 0.01 (two-tailed Student’s *t*-test).

## Data Availability

The data presented in this study will be openly available in GEO, reference number GSE171117. To review GEO accession GSE171117, go to https://www.ncbi.nlm.nih.gov/geo/query/acc.cgi?acc=GSE171117 and enter token mtsdkiayfvsdnep into the box.

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
