# Peer review of "Identification and Functional Characterization of Novel MYC-Regulated Long Noncoding RNAs in Group 3 Medulloblastoma"

_cancers, 2021, doi:10.3390/cancers13153853_

Round 1

Reviewer 1 Report

Authors have done a really thorough job characterizing the 3 lnc's regulating MYC and have also uncovered mechanisms important to each. They should be commended for their diligence. I have 2 minor suggestions prior to publication:

Introduction

"However, although displaying similarities with Group 4 subgroup, MYC signature is restricted to G3: it occurs in about 17% 61 of patients whereas it is extremely rare in other MB subgroups [2], [6]." c-MYC is restricted to G3MB; n-MYC is found in G4MB (group 4 alpha). Refer to Cavali et al. 2017. Please correct this statement. 

Results

"Derived from MYC-driven primary MB tumors, it is presently regarded as a cell system at the crossroad between Group3 and Group 4 MB, reflecting in vitro the partial overlap observed between these tumor sub-groups in vivo [30]. Importantly, in the framework of this study, D283 MED cells display an amplification of MYC expression [30], [31], which makes them an ideal cell model where investigating how MB transcriptome responds to perturbation of MYC activity.Given your primary choice of cell line for this study, i.e. D283, which is a G3/4 cell line, is it advisable to draw a strong distinction between G3 and G4 differences with regards to MYC, given the overlap? Both do express MYC, c- vs. n- and both have different aggression profiles based on this expression. The  best cell line to use for G3MB c-MYC amplification would have been D341 or HDMB03. I am not suggesting you re-do all your good work but consider how to justify the use of a G3/4 MB cell line when your primary point is that c-MYC is the aggressive trigger in G3MB. This comment relates to the prior comment about MYC being restricted to G3 tumors --> why then use a G3/4 tumor cell line as opposed to a purely G3M cell line for your studies? 

Author Response

  1. "However, although displaying similarities with Group 4 subgroup, MYC signature is restricted to G3: it occurs in about 17% 61 of patients whereas it is extremely rare in other MB subgroups [2], [6]." c-MYC is restricted to G3MB; n-MYC is found in G4MB (group 4 alpha). Refer to Cavali et al. 2017. Please correct this statement.

According to the reviewer’s suggestion, we have clarified in the Introduction that Group 3 and Group 4 MB are characterized by a c-MYC and n-MYC gene signatures, respectively.

  1. Derived from MYC-driven primary MB tumors, it is presently regarded as a cell system at the crossroad between Group3 and Group 4 MB, reflecting in vitro the partial overlap observed between these tumor sub-groups in vivo [30]. Importantly, in the framework of this study, D283 MED cells display an amplification of MYC expression [30], [31], which makes them an ideal cell model where investigating how MB transcriptome responds to perturbation of MYC activity." Given your primary choice of cell line for this study, i.e. D283, which is a G3/4 cell line, is it advisable to draw a strong distinction between G3 and G4 differences with regards to MYC, given the overlap? Both do express MYC, c- vs. n- and both have different aggression profiles based on this expression. The  best cell line to use for G3MB c-MYC amplification would have been D341 or HDMB03. I am not suggesting you re-do all your good work but consider how to justify the use of a G3/4 MB cell line when your primary point is that c-MYC is the aggressive trigger in G3MB. This comment relates to the prior comment about MYC being restricted to G3 tumors --> why then use a G3/4 tumor cell line as opposed to a purely G3M cell line for your studies

We appreciated the referee’s further suggestion, and we provide here additional clarifications underlying the choice of the long established and largely used D283MED cell line as an in vitro system for modeling MB in our study.

With regard to the fact that D283 MED cells are considered at the crossroad between Group 3 and Group 4 MB, it is important to underline, in the framework of this study, that D283 MED cells overexpress the Group 3 driver gene c-MYC and not its paralog n-MYC, that is a driver gene of Group 4 (see the RNA-seq Dataset S1, now introduced in the manuscript). The overexpression of c-Myc in D283 MED cells makes them a reliable model where analyzing c-MYC dependent molecular circuits.

In more details, the choice of D283 MED cells was based on the following criteria: we have established by qRT-PCR analysis and shown in the new Fig. S1, that the c-MYC gene is overexpressed at comparable levels in both the D283 MED and D341 cells. However, D283 MED cells have been chosen over the D341 cells for their consistently shorter doubling timing, that makes them, in our hands, more suitable for reverse genetics approaches aimed to highlight lncRNA function.

On the other hand, the other Group3-derived HD-MB03 cell line showed much higher levels of c-MYC expression compared to D283 MED, which could make less effective the functional inhibition of c-MYC activity through the OMOMYC strategy.

Nevertheless, the D341 and HD-MB03 cell lines were exploited to validate the upregulation of the three lncRNAs with respect to normal cerebella (Fig. S5).

We have introduced these comments and data in the text.

Reviewer 2 Report

Authors have addressed my comments in the revised version.

Author Response

We appreciate that our answers and new data satisfied your concerns.

This manuscript is a resubmission of an earlier submission. The following is a list of the peer review reports and author responses from that submission.

Round 1

Reviewer 1 Report

In this study, authors study group 3 medulloblastomas, working to unveil their molecular basis, by focusing specifically on Myc-regulated long non-coding RNAs, whose upregulation in group 3 tumors serves as a marker of poor prognosis and elevated tumorigenicity. By stably inhibiting Myc in D283 (Myc+) cell line (Group 3/4), they employ an initial in silico approach to reveal multiple up- and down-regulated Myc-dependent genes. In doing so, authors identified AC116407.1 (lncMB1), AC091182.1 (lncMB2), and AC010998.3 (lncMB3) and studied these in further detail, linking each to group 3 MB tumorigenicity. Overall, the strengths of the manuscript are the strong premise and rationale for studying Myc in group 3 MB; use of a stable Myc inhibition model for their subsequent in silico analysis to identify plausible myc-regulated targets; and the thorough in silico analysis done.

However, the study does falls short in the characterization of these lnc's.

  1. Authors did a nice job identifying their targets. However, once identified, the first task must be validating their choices ex vivo and in vitro. As they have done in Figure 6 (in tumors), authors should present the expression of their chosen lnc in vitro in several group 3 MB cell lines (D283, D341, D425, HDMB03) compared to control (normal human astrocyte or normal progenitor cells) to demonstrate that these lnc are in fact highly expressed in myc-driven cancer cells. This will increase rigor for the findings. 
  2. Figure 6 might be better presented as a heat map of all the down regulated genes isolated by your in silico analysis followed by the expression graphs presented of the selected ones. Consider interrogating these lnc's in the Weishaupt et al. Bioinformatics 2009 dataset that pooled samples from multiple prior studies to see if these trends are significant (place in supplement for further confidence in discovered trends; might run into issues with study heterogeneity which may not provide as high confidence as the single dataset used, but worth looking into if it can strengthen your claims). 
  3. Also, since these are purported to be only myc regulated, presumably in SHH tumors, these transcripts would not be affected. Can show this in tumors using the Kanchan et al. or Weishaupt et al. online cohorts and using the DAOY cell line (SHH type). 
  4. Can authors associate high expression of these lnc with poor survival? Consult R2database (https://hgserver1.amc.nl/cgi-bin/r2/main.cgi), Cavali MB dataset. 
  5. Why didn't authors include AL022313.4, AP005901.5, ACKR4P1, TRIM51FP, LINC02754, AP005901.5, AC112236.3, or AC069222.1, all of which were only affected by Myc inhibition alone and not Dox induction? Better rationale for choosing the 3 lnc should be presented. 
  6. Myc amplification is a cardinal high-risk feature of group 3 MB, and its phosphorylation status influences downstream tumor phenotype. More specifically, Myc phosphorylated at serine 62 increases stability leading to tumor aggressiveness, while phosphorylation at threonine 58 destabilizes the protein, leading to ubiquitin-mediated degradation and subsequent cellular apoptosis. Authors could show what happens to differential Myc phosphorylation to determine if any of these lnc form an autocrine loop for ongoing Myc-driven tumorigenicity. 
  7. For completeness, in studying caspace-dependent cell death, authors should also show the effect of lnc on caspases 3 and 9. 
  8. Aside from viability, what about colony formation, migration/invasion, wound healing - all common cancer cell assays that are simple to perform and can provide support for the neoplastic potential of the lnc's being studied. 

Minor points to address:

Introduction

"...and the worst outcome at 5 years (under 60%) [4], [5]." Please indicate under 60% refers to what specifically? 5-year survival? Event-free survival?  

Results

Line 418: "...did not caused..." please correct grammar 

Reviewer 2 Report

The manuscript by Jessica Rea et al presented an interesting finding where they identified MYC dependent lncRNAs in Medulloblastoma.

Overall authors have interesting finding and experimental procedure is clearly depicted which makes the manuscript easy to read and presentation of the data is also clear.

I have 2 comments which I feel could improve the finding further:

  1. Are some of the MYC dependent lncRNAs are bound by MYC directly, chIP-qPCR validation of few of them will provide further insight if they were under direct regulator of MYC
  2. the clinical connection of the MYC dependent lncRNAs I find a bit weak. Are these lncRNAs were specifically having high expression in group 3 or also in other group as well. A comparison (with other group) is required and that is one of main goal of the studies as depicted in introduction. More of the lncRNAs should be included in this comparison analysis across subgroup. the analysis. the argument why other lncRNAs were not selected is also not clear. Were the functionally studied one were most clinically relevant. To me these descriptions are required to make the observation clinically connected which is one of goal of the studies described by authors in abstract and introduction.
